# Raman Spectroscopy: A Personalized Decision-Making Tool on Clinicians’ Hands for In Situ Cancer Diagnosis and Surgery Guidance

**DOI:** 10.3390/cancers14051144

**Published:** 2022-02-23

**Authors:** Maria Anthi Kouri, Ellas Spyratou, Maria Karnachoriti, Dimitris Kalatzis, Nikolaos Danias, Nikolaos Arkadopoulos, Ioannis Seimenis, Yannis S. Raptis, Athanassios G. Kontos, Efstathios P. Efstathopoulos

**Affiliations:** 1Department of Medicine, Democritus University of Thrace, 68100 Alexandroupolis, Greece; mariakouri90@gmail.com (M.A.K.); spyratouellas@gmail.com (E.S.); mariakarnach@gmail.com (M.K.); 22nd Department of Radiology, Medical School, National and Kapodistrian University of Athens, 11527 Athens, Greece; dimitriskalatzis@gmail.com; 3Medical Physics Program, Department of Physics and Applied Physics, Kennedy College of Sciences, University of Massachusetts Lowell, 265 Riverside Street, Lowell, MA 01854, USA; 4Physics Department, School of Applied Mathematical and Physical Sciences, National Technical University of Athens, Iroon Politechniou 9, 15780 Athens, Greece; yraptis@mail.ntua.gr (Y.S.R.); akontos@mail.ntua.gr (A.G.K.); 54th Department of Surgery, School of Medicine, Attikon University Hospital, University of Athens, 1 Rimini Street, 12462 Athens, Greece; ndanias@med.uoa.gr (N.D.); narkadopoulos@med.uoa.gr (N.A.); 6Medical School, National and Kapodistrian University of Athens, 75 Mikras Assias Street, 11527 Athens, Greece; iseimen@med.uoa.gr

**Keywords:** Raman spectroscopy, cancer, diagnosis, in situ biopsy, surgical oncology, molecular fingerprint

## Abstract

**Simple Summary:**

Cancer still constitutes one of the main global health challenges. Novel approaches towards understanding the molecular composition of the disease can be employed as adjuvant tools to current oncological applications. Raman spectroscopy has been contemplated and pursued to serve as a noninvasive, real time, in vivo tool which may uncover the molecular basis of cancer and simultaneously offer high specificity, sensitivity, and multiplexing capacity, as well as high spatial and temporal resolution. In this review, the potential impact of Spontaneous Raman spectroscopy in clinical applications related to cancer diagnosis and surgical removal is analyzed. Moreover, the coupling of Raman systems with modern instrumentation and machine learning methods has been explored as a prominent enhancement factor towards a personalized approach promoting objectivity and accuracy in surgical oncology.

**Abstract:**

Accurate in situ diagnosis and optimal surgical removal of a malignancy constitute key elements in reducing cancer-related morbidity and mortality. In surgical oncology, the accurate discrimination between healthy and cancerous tissues is critical for the postoperative care of the patient. Conventional imaging techniques have attempted to serve as adjuvant tools for in situ biopsy and surgery guidance. However, no single imaging modality has been proven sufficient in terms of specificity, sensitivity, multiplexing capacity, spatial and temporal resolution. Moreover, most techniques are unable to provide information regarding the molecular tissue composition. In this review, we highlight the potential of Raman spectroscopy as a spectroscopic technique with high detection sensitivity and spatial resolution for distinguishing healthy from malignant margins in microscopic scale and in real time. A Raman spectrum constitutes an intrinsic “molecular finger-print” of the tissue and any biochemical alteration related to inflammatory or cancerous tissue state is reflected on its Raman spectral fingerprint. Nowadays, advanced Raman systems coupled with modern instrumentation devices and machine learning methods are entering the clinical arena as adjunct tools towards personalized and optimized efficacy in surgical oncology.

## 1. Introduction

According to World Health Organization (WHO) [1], in 2020 nearly 10 million cancer deaths have been accounted worldwide while the most common cancer cases pertain breast cancer (2.26 million cases); lung cancer (2.21 million cases); and colon and rectum cancer (1.93 million cases) [2,3,4]. Therefore, the early and accurate diagnosis as well as the precise and adequate surgical removal of a malignancy can lead to the reduction of cancer’s high mortality rates [5,6,7]. Since the differentiation among benign tumors, premalignant, early-stage malignant and healthy tissue is challenging, repeated biopsies are often necessary. Positive predictive values regarding tissue sampling are as low as 22% for prostate cancer diagnosis, 1.4% for breast cancer, 18.5% in lung cancer screenings and 7–23% for melanoma diagnosis [8,9,10,11]. Various conventional imaging techniques have attempted to serve as adjuvant tools for biopsy and surgery guidance. In the field of ionizing radiation, positron emission tomography (PET), computed tomography (CT) and single photon emission computed tomography (SPECT) offer great results, with undisputable drawback relating to the dose deposition to the patient [12,13,14]. Simultaneously, magnetic resonance imaging (MRI), optical coherence tomography (OCT), white light reflectance (WLR), fluorescence, and high frequency ultrasound by exploiting non-ionizing radiation have proved to be valuable diagnostic tools [15,16,17,18]. Nonetheless, currently, no single imaging modality has been proven sufficient in terms of the required standards of specificity, sensitivity, multiplexing capacity, spatial and temporal resolution, and low cost [19,20]. Moreover, most techniques are unable to provide information regarding the molecular tissue composition [21,22]. They just confide on visual changes of the tissue structure and thus present lack of specificity [23,24].

Optical vibrational spectroscopic techniques, such as Raman spectroscopy (RS), can depict the gradual changes among malignant and healthy tissue by exploiting the analysis of the characteristic Raman spectrum [24,25]. Raman spectroscopy is a spectroscopic technique offering high detection sensitivity and spatial resolution of a few μm. In general, RS provides information of the short-range molecular vibrational level where the Raman bands are characteristic of the molecular bonding in each chemical group. Eventually, the chemical conformation and the environment of the macromolecular level determine the exact frequencies of the Raman bands. Therefore, Raman Scattering can provide exquisite detail of particular sites of interest and of any biochemical alteration related to the inflammatory, or cancer state of tissue. These tissue related details are reflected on the spectral fingerprint [26] since Raman spectrum constitutes an intrinsic “molecular fingerprint” of the sample [27,28]. This leads to a treasure of information regarding the vibrational modes related to specific chemical configurations present in tissues, correlated with proteins, lipids, glucose consumption, DNA, RNA, and other biomolecules [29]. Consequently, the entirety of a Raman spectrum can provide the analytic guideline of biological sample’s structure, identity, and composition as well as the depiction of macromolecules interactions and composition [27]. Due to that, RS offers high molecular specificity into the characterization of biological tissues ex vivo and in vitro and constitutes an excellent non-invasive detection method of the molecular differences among tumor and healthy tissue [30,31,32]. Moreover, RS does not require any reagents, labelling or other preparation of the tissue while the use of optical fibers allows the assessment of several anatomical locations in vivo [21,31,33,34].

Vigorous attempts have been made during the last decade towards the clinical implementation of Raman spectroscopy in the hope of addressing the same fundamental issue: the inadequacy of pre- and intraoperative methods with satisfactory and clinically relevant specificity and sensitivity. The most recent studies are aiming into the quadruple of: (1) pre-malignant lesions detection, (2) detection of cancer in less advanced stages, (3) the reduction of unnecessary biopsies, and (4) guided surgery for the entire removal of a malignancy with adequate tumor resection margins [35,36,37,38]. According to the literature, in vivo and ex vivo trials which are aiming towards the detection of malignant tissue have accomplished specificities varying between 45–100% and sensitivities varying between 77–100% [39,40,41,42]. Studies with the goal of pre-malignant lesions detection observed sensitivities ranging among 70–93.5% and specificities ranging between 63–97.8% [43]. Even though the numbers presented may not touch perfection, they still constitute a strong argument towards the capability of Raman spectroscopy to enhance current clinical practice.

In this review, we provide an overview of the most prominent Raman spectroscopy applications in biological and clinical research. We highlight the perspective of advanced Raman systems incorporation in clinical praxis as an adjunct tool towards early diagnosis and oncologic surgery guidance. The combination of Raman spectroscopy with modern instrumentation devices, novel techniques, and machine learning methods is presented. This coupling will contribute to overcoming current limitations which have prevented the broad clinical application of Raman spectroscopy so far and will establish RS’s potential to be used as a personalized decision-making tool.

## 2. Raman Spectra Analysis for Tissue Characterization

In general, configurations of nearby chemical bonds are characterized by typical vibrational energies. When photons, emitted by a laser light source, are inelastically scattered by these characteristic molecular oscillations, a Raman scattering event takes place. The detection and analysis of the scattered photons offers a spectrum comparted of the so called characteristic Raman peaks. Each individual peak is indicative of a particular vibrational mode related to distinct chemical configurations [22,25]. Various Raman techniques have been developed to cover the distinct requirements of each biomedical sample such as: Spontaneous Raman Spectroscopy (SRS) [27,44,45], Resonance Raman Spectroscopy (RRS)[45], Surface-Enhanced Raman Scattering (SERS) [46,47,48,49,50], and Coherent anti-Stokes Raman scattering (CARS) [51,52,53,54] etc. Nevertheless, despite the advancement of these techniques, they still present complexities during experimentation and analysis and thus cannot be yet applied as a simple surgery tool. This review will be concentrated on conventional Spontaneous RS. 

Prominent tissue Raman peaks are observed in the fingerprint of 800–1800 cm^−1^ and the high frequencies 2800–3200 cm^−1^, spectral regions. A characteristic example of Raman spectra could be this from colorectal tissues by Bergholt et al. Τhey performed discrimination between normal, hyperplastic, adenoma, and adenocarcinoma using near-infrared Raman spectroscopy [55]. Figure 1 shows the mean of in vivo Raman spectra of normal (*n* = 1464), hyperplastic polyps (*n* = 118), adenoma (*n* = 184), and adenocarcinoma (*n* = 103) acquired from 121 lesions of 50 patients during colorectal endoscopy. The strongest Raman bands are marked upon the spectra and are related to specific vibrations in cellular or extracellular components: 853 cm^−1^ (ν(C–C) proteins), 1004 cm^−1^ (νs(C–C) ring breathing of phenylalanine), 1078 cm^−1^ (ν(C–C) of lipids), 1265 cm^−1^ (amide III ν(C–N) and δ(N–H) of proteins), 1302 cm^−1^ (CH_2_ twisting and wagging of lipids), 1445 cm^−1^ (δ(CH_2_) deformation of proteins and lipids), 1618 cm^−1^ (ν(C=C) of porphyrins), 1655 cm^−1^ (amide I ν(C=O) of proteins), 2850 cm^−1^ and 2885 cm^−1^ (symmetric and asymmetric CH_2_ stretching of lipids), 2940 cm^−1^ (CH_3_ stretching of proteins), 3009 cm^−1^ (asymmetric = CH stretching of lipids). Bands above 3200 cm^−1^ are OH stretching modes of water.

## 3. Machine Learning and Deep Learning as Tools towards Raman Spectra Analysis

The analysis of the vast amount of Raman data is a critical barrier which needs to be overcome in order to enable the facilitation of Raman spectroscopy in the clinical routine [56]. However, the evolution of artificial intelligence (AI) provided a boost in real-time Raman data processing. The combination of AI tools with Raman spectroscopy can efficiently lead to adequate discrimination of cancerous tissues [57]. Machine learning (ML) and Deep learning (DL) constitute branches of the broader division of Artificial Intelligence (AI) [58]. Their advanced innovation deservedly classifies them as an excellent candidate for medical applications especially for those dependent on complex, highly versatile genomic procedures such as cancer diagnosis and detection [59,60,61]. ML or DL could constitute valuable tools in physics applications in medicine such as Raman spectroscopy, where the detection and analysis of various spectral fingerprints is vital [62,63,64]. 

Figure 2 is a schematic representation of the workflow of the combination of Raman spectroscopy with machine learning models for tissue discrimination and classification using a multilayer perceptron (MLP).

In more detail, features extracted from Raman spectra (e.g., representative intensities at certain wavenumbers) and/or from spectral images of biostructures (e.g., pixel intensity patterns) are used as inputs to an MLP. The hidden layers of an MLP will introduce a series of linear and non-linear calculations that will lead to a single output neuron. Each output corresponds to normal or malignant class towards the discrimination between healthy and cancerous tissues. Several machine learning models such as Support Vector Machine (SVM) [65], boosted tress [66], convolutional neural networks (CNNs) [67], and artificial neural networks (ANNs) [68] have been exploited for cancer detection for almost 20 years [69]. 

Xiaozhou Li et al. focused on the expediency of Raman spectroscopy for colon cancer diagnosis by serum analysis. They observed statistically important spectroscopic differences among cancerous and normal cells for six Raman peaks at 750, 1083, 1165, 1321, 1629, and 1779 cm^−1^, which indicate nucleic acids, amino acids, and chromophores respectively. The intensity of the peaks in the cancerous cells either increases or decreases reflecting the induced chemical modifications. For example, the decrease of the 1165 cm^−1^ peak is related to low levels of anti-oxidant β-carotene in the cancerous cells. They used principal component analysis (PCA) and k nearest neighbor analysis (KNN). They concluded that a number of the PC loading peaks are identified as colon tissue peaks which eventually proved the correlation among the original Raman spectra and the PC loading spectra. Specificity calculated by KNN analysis reached 92.6% and a diagnostic accuracy of 91% [70]. The diagnostic models built with the identified Raman bands provided diagnostic accuracy of 93.2% into identifying colorectal cancer.

Ragini Kothari et al. investigated rapid, quantitative, probabilistic breast tumor assessment with real time error analysis. They observed that often the spectral shifts that were denoted as malignant would constitute false positives due to lack of lipid signals [71]. Stochastic neural networks (NNs) were exploited to estimate the Bayesian probability of a Raman spectrum containing characteristic peaks of cancer using data from the entire spectral bandwidth (600–3000 cm^−1^), the fingerprint region (600–1800 cm^−1^), and the high wavenumber region (2800–3000 cm^−1^) [68]. Qingbo Li et al. suggested an entropy weighted local-hyperplane k-nearest-neighbor (EWHK) algorithm to determine the Raman spectra in breast cancer by enhancing the classification accuracy [72]. This method led to a positive prediction rate of 95.99%, a negative prediction rate of 83.69%, specificity of 87.77%, accuracy of 92.33%, and sensitivity of 93.81% [72].

Shaoxin Li et al. used near-infrared Raman spectroscopy and feature selection approaches to detect colorectal cancer tissues. Significant differences were identified between normal and cancerous cells by using ant colony optimization (ACO) and support vector machine (SVM) algorithms for five Raman bands related to proteins, nucleic acids, and lipids of tissues in the areas of 815–830, 935–945, 1131–1141, 1447–1457, and 1665–1675 cm^−1^. For example, the 1323 cm^−1^ band, which is assigned to nucleic acids (CH_3_CH_2_ twisting mode), increases in cancer tissues compared to normal ones, reflecting the higher content of nucleic acid in tumor cells. A diagnostic accuracy of 93.2%, a sensitivity of 92.3%, and a specificity of 94.2% were achieved [73].

Non-linear NNs have been used to predict the Bayesian probability of breast cancer. Nine spectra regions, six in the fingerprint region (600–1800 cm^−1^) and three in the high wavenumber region (2800–3200 cm^−1^), were identified comparing DNA/RNA, protein, carbohydrate, and lipid cellular components of healthy and cancerous cells [71]. Deep convolution neural networks have been applied to fiber optic Raman spectroscopy systems providing a novel classification method for tongue squamous cell discrimination [74] According to the results, high sensitivity (99.31%) and specificity (94.44%) were achieved. 

## 4. Advanced Raman Systems in Clinical Praxis

### 4.1. Raman Systems for Early Diagnosis

According to the literature, Raman spectroscopy-based biopsy guidance presents overall specificities and sensitivities between 66–100% and 73–100% respectively [21,24]. The use of this technique promises a drastic increase in the accuracy of cancer diagnosis and an important reduction in the number of false positive biopsies [31,33,34]. The detector technology improvement, the in vivo fiber-optic probe design and the use of artificial intelligence algorithms as well as the collection of large independent comprehensive datasets obtained in the actual clinical workflow enable the facilitation of Raman-based systems into the routine clinical settings [23].

Fiber-optic probes have enabled the access of Raman spectroscopy in in vivo diagnostic techniques [55]. The ability of fiber probes to be inserted endoscopically, especially in hollow and solid organs, such as the upper gastrointestinal tract, the colorectal, and cervical cancers, or the oral cavity, the bladder, and the lung, enables in vivo measurements and in vivo assessment [55,75,76,77]. Advanced fiber probes such as probes with plasmonic nanostructures on their distal end surface can provide enhancement of the surface Raman scattering signal [78]. Moreover, fiber probes can overcome the limited penetration depth of laser radiation in tissues due to the high diffusion and scattering of photons. Figure 3 shows a portable Raman imaging system based on SERS fiber-optics probes capable of conducting white light endoscopy [79].

Moreover, novel Raman techniques combined with advanced fiber probes can offer a boost to Raman Spectroscopy’s application in clinical praxis. For example, Micro-scale spatially offset Raman spectroscopy with an optical fiber probe (micro-SORS) can collect photons from deeper layers by offsetting the position of the laser excitation beam [80] and by reaching a penetration depth up to 5 cm [81]. Recently, Zhang et al. combined micro-SORS with Surface-enhanced Raman spectroscopy (SERS) applied on a tissue phantom of agarose gel and biological tissue of porcine muscle [82]. According to their results, the penetration depth could be improved over 4 cm in agarose gel and 5 mm in porcine tissue compared to the 2 cm depth of agarose gel and the 3 mm depth in porcine muscle received by SERS measurements.

Stevens et al. and Wang et al. investigated epithelial tissue associated with dysplasia and developed a Raman probe coupled with a ball lens that could enhance in vivo Raman measurements from gastric premalignant epithelial tissue during endoscopy [83,84]. Due to the use of a ball lens, they managed to decrease the collection depth at 300 nm, which is the relevant depth for the analysis of gastric epithelium [83,84]. Moreover, they exploited a multimodal image-guided Raman technique to achieve real time in vivo cancer detection. Bergholt et al. used this high wavenumber system in combination with a foot pedal control switch and auditory feedback to the gastroenterologist during colonoscopy diagnosis [85]. Another team, Agenant et al., developed a novel Raman probe that could take measurements at the depth of 0–200 μm (average urothelium depth), the adequate level for superficial tissue sampling, in order to improve in vivo diagnosis of urothelial carcinoma [86]. This novel probe was comparted of seven collection fibers, one excitation fiber and two component front lens [86]. Figure 4 shows the different geometries of fiber-optics probes used in clinical applications such as endoscopic probes without any focusing optics, confocal endoscopic probes, and fiber probes for side-viewing [87]. 

Challenges of fiber probe’s use pertain to the intense resemblance between the excitation laser light and the collection of the scattered light by the different tissue’s anatomical regions [88]. Some of the problems arise due to the background Raman and fluorescence signals created by the fiber’s materials and due to the intrinsic fluorescence signal (autofluorescence) of the tissue [54,89]. The separation between the collection and the excitation pathways is still a valid issue for Raman tissue measurements. The background Raman and fluorescence signals created inside the fiber require the separation between the collection and excitation pathway [27]. This generates a challenge regarding the minimum size of such devices. Nijssen et al. attempted to overcome this difficulty by detecting the high wavenumber region from 2500 to 3800 cm^−1^ (near-infrared region) of the Raman spectrum [47]. That way, the same silica-based fiber optic probe could both guide laser light to the tissue and simultaneously collect scattered light. At the same time, low overlap was achieved with the generated parasitic signals (Rayleigh scattering and Raman from the probe) [47] offering that way a perspective towards the miniaturization of such systems.

Moreover, the development of Raman instrumentation regarding in vivo and ex vivo applications is mainly focusing on the overcoming of issues such as: the speed of measurement, the instrumentation cost, and the background interference due to the different types of tissue. Advanced focal-plane detectors, volume-phase holographic gratings, stabilized diode lasers, and imaging polychromators are building a new perspective towards robust Raman instrumentation [90,91] which achieves high quantum efficiency, simultaneous spectral integration from the high spectral and lateral range and low background noise [90,91]. Therefore, the traditional limitations of low sensitivity and poor detection capability that Raman spectroscopy systems used to present are now dropping drastically [92]. New innovative techniques allow infrared and near infrared detection while cutting edge technologies promise system architectures with single photon detection capabilities and hybrid imaging technologies [91,93,94]. The development of an in vertical-external-cavity surface-emitting semiconducting laser presents a large gain area and transverse mode control of the extended cavity, and hence accomplishes a combination of high continuous wave output power and a near diffraction limited beam [93]. Furthermore, semiconductor lasers present the advantages of easy array fabrications and low cost of production [95]. 

### 4.2. Raman Systems for Guided Surgery

According to current practice, the primary treatment for solid tumors is surgical removal [96,97,98,99]. Adequate surgical margins, vital for disease control, are selected for the resection of the entirety of the cancerous tissue. Of vital importance is the preservation of all healthy structures, in cases where the anatomical regions allow it. However, the surgical resection techniques that are currently used are based on subjective methods, such as visual inspection or palpation to verify the exact margins between malignant and normal tissue [96,97,98,99]. This may lead to partial removal of the malignancies and consequently to the occurrence of residual tumors, strongly correlated with poor survival rates [96,97,98,99]. In addition to that, additional surgeries, or adjuvant therapies such as radiation therapy or chemotherapy may be required. Studies indicate that the five-year survival rate decreases drastically when a solid tumor is not dissected to its entirety [23,96,97,98,99]. Portable Raman systems have been implemented into the clinical environment of oncological surgeries presenting excellent assets such as the ability to offer representative sampling towards correct pathological diagnosis and accurate assistance in the definition of resection margins during surgery. As can be depicted in Figure 5, the objectivity of Raman spectroscopy as an imaging technique collaborated with the data analysis and classification capabilities of Machine Learning techniques could constitute a valuable intraoperative guidance tool.

An intraoperative Raman system that directly measures brain tissue in patients has proven to distinguish dense and low-density cancer infiltration from benign brain tissue with a sensitivity of 93% and a specificity of 91% [100,101]. More precisely, the experimental setup was pertaining to a hand-held optic Raman probe and a 785 nm NIR Laser [101]. The research team exploited the boosted trees supervised machine learning algorithm to process their data and eventually differentiate the spectrum among cancerous and healthy brain tissue [101]. In another study, a real-time Raman intraoperative system was used during breast cancer surgery for the assessment of freshly resected specimens [102]. A total of 220 Raman spectra were collected with the aid of an 830-nm-diode laser focused on a Raman optical fiber probe [102]. This study has demonstrated that Raman spectroscopy could discriminate cancerous tissue from normal breast tissue with a sensitivity of 83% and a specificity of 93% [102].

A handheld contact Raman spectroscopy probe was used for real-time identification of brain cancer during surgery. Jermyn et al. obtained very fast and high-quality pure Raman signals from 0.5 mm tissue areas with sampling depth up to 1 mm during the tumor resection [66] by using micrometer-scale filters that were placed directly at the tip of the optical fibers [66]. A portable clinical fiber probe system in combination with a classification AI algorithm with the ability to differentiate healthy breast tissue from cancerous tissue was utilized by Barman et al. as a guidance tool for mastectomy procedures. The recorded specificity was 100% with sensitivity of 62.5% [33]. The differentiation among normal, breast cancer, fibroadenoma, and fibrocystic change was achieved with accuracy of 82.2% [33]. 

In order to reduce the time measurement of whole tissue sections in skin cancer, Kong et al. developed the approach of using auto-fluorescence images at excitation wavelengths of 377 nm and 292 nm in combination with Raman spectroscopy [42]. Since these wavelengths are the corresponding excited peaks of tryptophan and collagen, they managed to differentiate normal dermis (characterized by high collagen expression) to cancerous segments [42]. This method recorded measurements with specificity of 94% and sensitivity of 95% [42]. Short et al. conducted a study using Raman spectroscopy on ex vivo colon tissue from 18 patients, measuring both the fingerprint and high-wavenumber spectral regions [77]. The results indicated that, using the high-wavenumber region, the non-malignant and the malignant groups could be classified correctly with a specificity of 89% [77]. The authors referred that the high-wavenumber region could be used in vivo to improve the identification of neoplastic lesions. In the domain of colorectal cancer, Bergholt et al. using an endoscopic multi-fiber Raman probe measured both the fingerprint and high-wavenumber spectral regions of 50 patients in vivo [55]. The team attempted to differentiate Adenomatous polyps from hyperplastic polyps with a specificity of 83% and a sensitivity of 91% [55]. Table 1 presents an overview of in vivo Raman measurements for clinical applications that have been attempted for a variety of cancer types.

## 5. Challenges and Future Perspectives

The main advantages of RS, such as (a) its non-invasive character and compatibility with tissue physiology due to the weak water signal, (b) its suitability for in vivo fiber-optic applications on versatile cancers, and (c) its high specificity with simultaneous chemical analysis of the malignant tissues have been thoroughly described in the previous sections. There are, however, important limitations of the technique which hinder its establishment in the clinical setting. The most prominent constraints are the following: (a) the weak Raman signals which require long acquisition times, (b) the strong autofluorescence background which affects the quality of the acquired spectra hiding the weak Raman features and complicating the analysis, (c) the potential damage of the tissues by laser heating which is a rather complicated effect depending on the laser excitation wavelength and power, as well as the light absorption coefficient of the tissues, and (d) the subtle differences in the spectra which require sophisticated analysis [147]. Nevertheless, technical advancements in current generation Raman spectrometers and integration of machine learning techniques for big data analysis and cancer classification gradually tilt the balance towards the application of RS as a rapid diagnostic tool in clinical praxis. It is worthwhile to mention that most of the portable Raman spectrometer manufacturers make their own cooperative research in the field and include such applications in their technical notes. Despite the above limitations, Raman Spectroscopy constitutes a very promising technique for in-situ cancer diagnosis. Since the achievement of adequate surgical margins is vital for disease control and survival, an intraoperative guidance tool such as Raman probes will significantly limit the subjective methods which surgical resection techniques are currently based on (such as palpation and visual inspection). However, the effort to reliably assess resection margins in surgical oncology also suffers from constrains inherent in oncological surgery, such as non-standardized practices and the impact of epithelial or other dysplacias at the margins, as well as differences in reporting the status of the surgical margin. With regard to RS applications on excised tissues, the method employed for the retrieval of sections from resection margins or bioptic samples and the issue of postresection tissue shrinkage introduce extra variability. Recent creation of multidisciplinary networks like ClirSpec, Raman4Clinics (EU COST Action BM1401) and EPIC are significantly narrowing the distance among scientists and clinicians [27]. These networks are actively pursuing the standardization of measurements and of the preparation of the samples, the creation of data analysis protocols, and the settlement of a basis for transferability.

## 6. Conclusions

The compelling recent developments of Raman instrumentation, including the new technologies and the reduction of the cost of lasers, holographic gratings, detectors, and Raman probes as well as the entrance of machine learning models in the analysis of the data have contributed towards the overcoming of some of the method’s deficiencies. Thus, they have inspired numerous research groups towards the elaboration of Raman based techniques with biomedical orientation. Among them, cancer prognosis and diagnosis is an excellent candidate, promising low invasive, in vivo and real time detection, and accurate molecular characterization. RS can achieve specificities of 45–100% and sensitivities of 77–100% in cancer diagnosis and malignancies detections. The objectiveness of RS and the ability to provide biochemical information in combination with the advancement of the Raman probes that can be integrated in endoscopes and provide spectroscopic images of the tissues may be the solution to the fundamental problem of the deficiencies of pre- and intraoperative methods with adequate and clinically relevant specificity and sensitivity. The prospect of adaptation of Raman spectroscopy into the clinical environment could finally provide surgeons with the assurance of the intraoperatively adequate resection margins while it could upgrade patient’s surgical outcome and simultaneously minimize the adjuvant therapies needed.

## Figures and Tables

**Figure 1 cancers-14-01144-f001:**
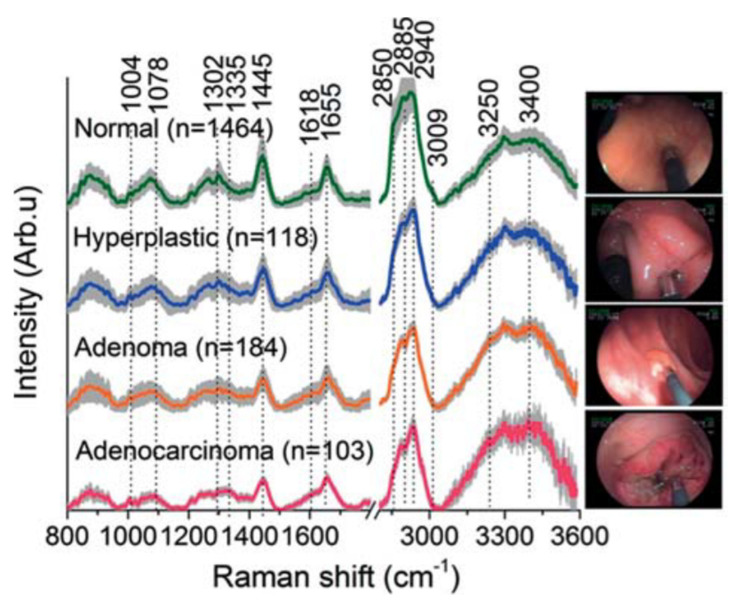
Mean ± 1 standard deviation (SD) values of in vivo fingerprint (FP) spectra (800–1800 cm^−1^) and high-wavenumber (HW) Raman spectra (2800–3600 cm^−1^) of normal (*n* = 1464), hyperplastic polyps (*n* = 118), adenoma (*n* = 184), and adenocarcinoma (*n* = 103) acquired from 121 lesions of 50 patients during colorectal endoscopy. The spectra have been normalized to the integrated area in the FP and HW ranges for comparison purpose. Reused with permission from [55]. Copyright 2015 WILEY-VCH Verlag GmbH & Co. KGaA, Weinheim, Germany.

**Figure 2 cancers-14-01144-f002:**
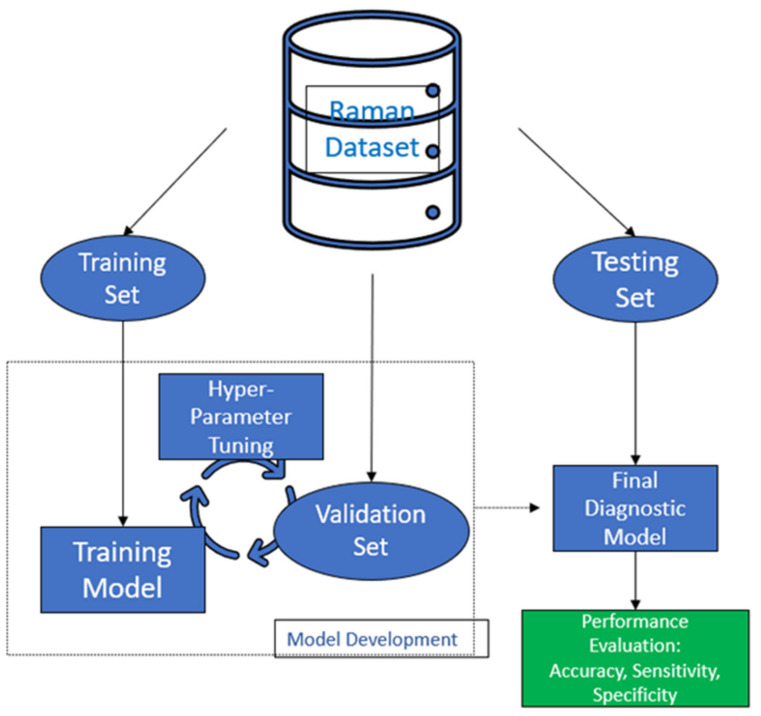
Depicts the basic structure of Machine Learning workflow applied on a Raman Dataset.

**Figure 3 cancers-14-01144-f003:**
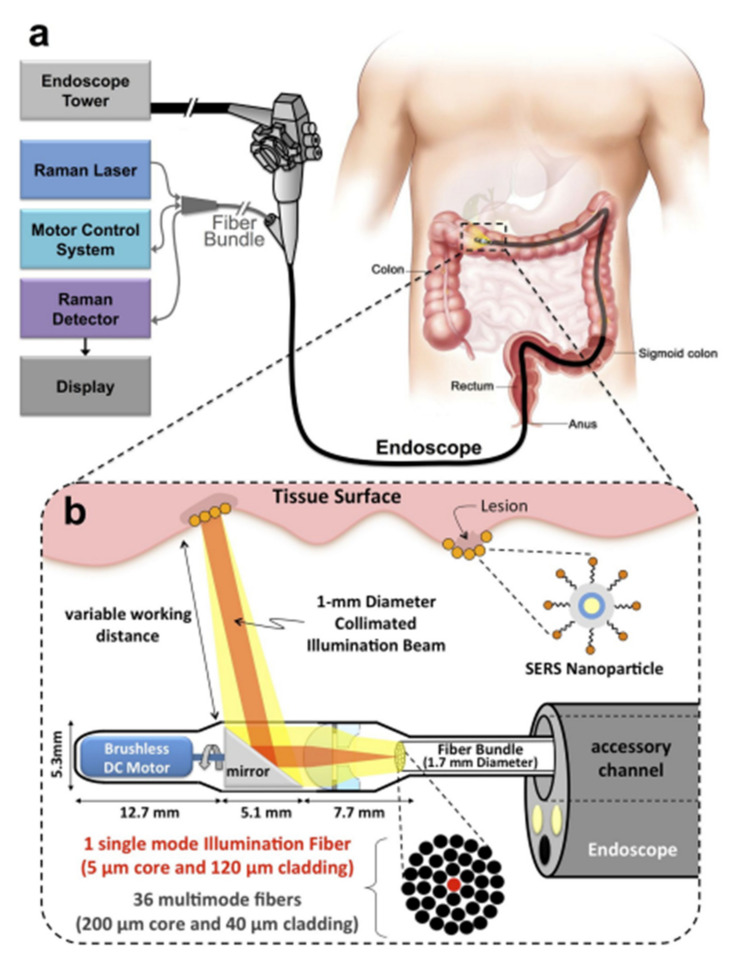
Schematic representation of a Raman system with SERS fiber-optic probe based Raman system which can perform white light endoscopy. (**a**) The design allows the Raman imaging system to get adapted on a clinical endoscope and scan the lumen as the endoscope is being retracted in the GI tract. (**b**) An expanded schematic illustration of the distal end of the device. The collimated beam can be swept by a brushless DC motor and its focus can be adjusted by a system of plano-convex and plano-concave lenses [79], https://doi.org/10.1371/journal.pone.0123185, access on 10 January 2022 ).

**Figure 4 cancers-14-01144-f004:**
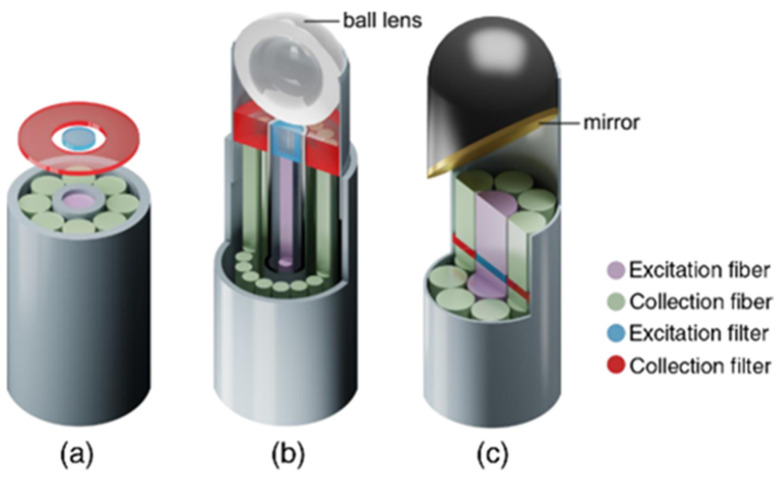
Different geometries of fiber-probes used in clinical applications: (**a**) non-superficial endoscopic probe with one excitation fiber in the center and seven collection fibers arranged around the emitter (**b**) confocal endoscopic fiber probe with a ball lens (**c**) fiber probes with mirror (or prisms) [87]. https://doi.org/10.1117/1.JBO.23.7.071210, access on 10 January 2022. PMID: 29956506. Excitation and collection filters are also depicted.

**Figure 5 cancers-14-01144-f005:**
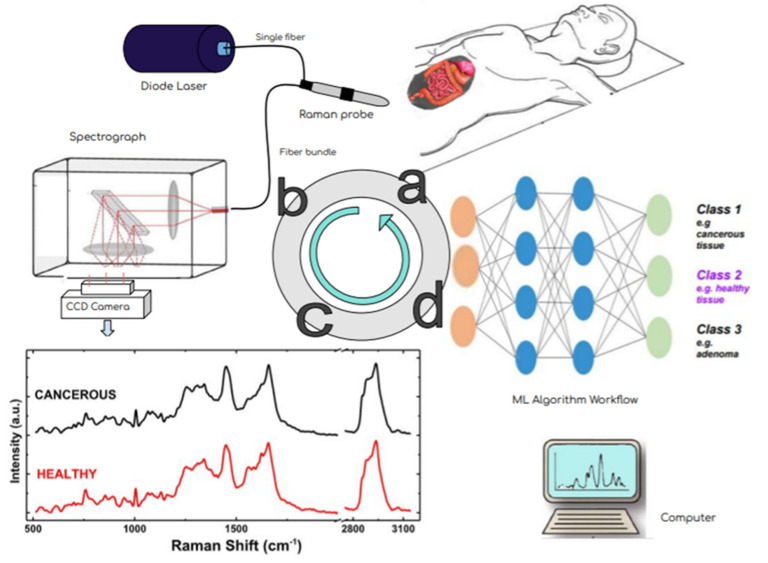
Effectiveness of oncologic surgery depends on precisely distinguishing healthy from malignant tissue during the operation. This flow diagram shows the steps of RS- based diagnosis from the patiant examination (**a**) via the multicomponent instrumentation (laser excitation–Raman probe-scattered light dispersion and detection) (**b**) in order to acquire the Raman spectra (schematic, not real data) in (**c**) and towards their analysis and classification via mashine learning techniques (**d**). A simple multi-layer perceptron neural network architecture is presented. In fact, the input layer is a data matrix with intensity values from different observations at various Raman frequencies. This combined methodology potentially has the ability to accurately differentiate benign from malignant tissue in real time and eventually improve the surgical outcome.

**Table 1 cancers-14-01144-t001:** Clinical Raman applications for diagnosis and surgery guidance.

Cancer Type	Current Practice (CP)	Accuracy (CP)	Raman Applications (RA)	Accuracy (RA)
Breast	Diagnosis Screening mammography [102]	*s: 72% *sp: 47% [102]	early diagnosis *RS tool for microcalcifications detection in breast tissue [103]	*ppv: 97% [103]
	Histopathological diagnosis: 1. Fine-needle aspiration cytology [104]	s: 82–99.7% [104]	surgery guidance *RS guided tool for mastectomy [33]	s: 62.5% sp: 100% *atd: 82.2% [33]
	2. Biopsy [105]	s: 90.1–93% [104]		
Skin	Diagnosis Visual inspection of morphologic characteristics with a dermoscope [106]	s: 68–96% [107] *fpsl: 30% [108] ppv: 7–23% [109]	early diagnosis Single fiber *Rp (in vivo): 1. Differentiation of malignant/benign lesions [110]	s: 91% sp: 75% [110]
			2. Distinguish of malignant melanoma/pigmented benign lesions [110].	s: 97% sp: 78% [110]
			3. Malignant/pre-malignant lesions separation from benign skin [110].	s: 90% sp: 64% [110]
			Portable R. system with handheld probe for non-melanoma skin/cancerous tissue [111].	s: 100% sp: 91% *ca: 95% [111]
			Multi-fiber R.p. (in vivo) for lesions clinically suspected of being skin cancer [112].	s: 52% sp: 67% [112]
			RS with auto-fluorescence for melanoma and BCC diagnosis [34,113].	*a: 97.3% [34,113]
			Surgery guidance RS (with auto-fluorescence) for intraoperative detection of BCC in skin [42].	s: 95% sp: 94% a: 85% [42,114]
Lung	Diagnosis White light bronchoscopy with tissue biopsy and cytologic evaluation [115,116,117].	s: low sp: low [115,116,117]	early diagnosis Multi-fiber RS probe (in vivo) for lesions with bronchoscopy [77,118].	s: 90% sp: 90% [77,118]
	Treatment Surgery (in early-stage disease) [119,120].	-		
Head and Neck	Diagnosis Screening and Biopsies [121]	-	early diagnosis Multi-fibre R.p. (in vivo) for malignant oral lesions classification [78].	s: 100% sp: 77% [78]
	Treatment 1. Surgery (early stage) [121,122]	*ss: 30–85% [121,122]	surgery guidance High-wavenumber R. spectra (ex vivo) for tumour identification [78]	- [78]
	2. Multimodality treatment: surgery, radiation, chemo/biotherapy, immunotherapy (advanced stage) [121,122]	*ss: 30–85% [121,122]		
			RS (ex vivo) for the borders of malignant/healthy tissue [78].	- [78]
Brain	Diagnosis 1. Neuroradiology [123,124,125,126]	-	surgery guidance Hand-held R. system probe (in vivo) for brain tumour resection- distinguishing normal brain tissue/dense cancer [127].	s: 93% sp: 91% [127]
	2. Stereotactic biopsy [123,124,125,126]	-		
	Treatment 1. Surgery [123,124,125,126]	-		
	2. Three-dimensional stereotactic navigation (5-ALA-fluorescence, MRI for surgical guidance [123,124,125,126]	-		
Colorectal	Diagnosis Screening by colonoscopy [128].	CRS *m.r: 2–6% [129,130]	early diagnosis RS (ex vivo) on colon tissue for non malignant/malignant group classification [131].	sp: 89% [131]
		adenomas *m.r: 20–26% [129,130]		
	Treatment Surgery for localized colon cancer [132,133,134]	-	Endoscopic multi-fibre R.p. (in vivo) for the separation of adenomatous polyps/hyperplastic polyps [135].	s: 91% sp: 83% [135]
Cervical	Diagnosis Screening by cervical cytology (test PAP) [136,137,138]	s: <50% sp: 95–98% [136,137,138]	early diagnosis High-wavenumber ball-lens fiber-optic RS probe (in vivo) for cervical pre-cancer diagnosis [139].	s: 94% sp: 98% [139]
	histopathology (colposcopy guided biopsy) [136,137,138]	s: 92% sp: 67% [136,137,138]	Portable fiber-optic R.p. for colposcopy-guided biopsy to investigate dysplasia [140].	s: 86% sp: 97% ca: 88% [140]
	Treatment 1. surgery (small tumours) 2. chemoradiation (higher stages)[141,142]	-		
Prostate	Diagnosis 1. transrectal ultrasound 2. guided prostatic biopsy [143]	-	Raman applications in real clinical area present difficulties due to limitations in research [86,144,145,146].	
	Treatment 1. radical prostatectomy 2. radiation therapy [143]	-		

*s: sensitivity, *sp: specificity, *ppv: positive predictive value, *atd: accuracy of tissue differentiation, *fpsl: false positive suspicious lesions, *ca: classification accuracy, *a: accuracy, *ss: surgical success, *m.r: miss rates, RS: Raman spectroscopy, R.p.: Raman probe.

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
