# Peer review of "Raman Spectroscopy: A Personalized Decision-Making Tool on Clinicians’ Hands for In Situ Cancer Diagnosis and Surgery Guidance"

_cancers, 2022, doi:10.3390/cancers14051144_

Round 1

Reviewer 1 Report

This mini review summarizes the progress of clinical application of Raman spectroscopy, especially in early diagnosis, surgy guidance and biopsy. This is a topic of great interest and high potential application. However, this manuscript can be improved significantly. Here is my suggestions.

  1. About the Raman band assignment for different biomolecules should be introduced at the beginning, which is of fundamental importance.
  2. The chemcal differences between tissues should be introduced to match the Raman spectroscopic differences.
  3. About the several Raman equipments specified for the specified bioapplication, the equipment details with figures should be presented.
  4. The disadvantage of Raman and the challenges towards real clinic usage should be discussed in more details.

Author Response

STATEMENT OF CORRECTIONS

                                                 REVISED MANUSCRIPT

Dear Editor,

In response to the reviewers´ comments, the manuscript has been significantly revised. All the corrections have been performed according to the Reviewers’ suggestions. The changes have been highlighted with yellow color in the text. Bellow follows the replies and respective answers to the reviewers’ comments and enquiries. All answers have been uploaded separately as well.

Yours sincerely,

Dr. Spyratou Ellas

                              Reviewer’s comments and General Corrections

Reviewer 1

Comment 1: About the Raman band assignment for different biomolecules should be introduced at the beginning, which is of fundamental importance.

Reply: The Raman band for the different biomolecules was added in lines 134-136 according to the numbering of the revised manuscript:

Lines 134-136: “Prominent tissue Raman peaks are observed in the fingerprint, 800-1800 cm-1 and the high frequency, 2800-3200 cm-1, spectral regions. As a characteristic example, we present Raman spectra from colorectal tissues by Bergholt et al”

Comment 2: The chemical differences between tissues should be introduced to match the Raman spectroscopic differences.

Reply:  The chemical differences between tissues matching the Raman spectroscopic differences were added in lines 140-153 and the text was modified accordingly:

Lines: 140-153: “The strongest Raman bands are marked upon the spectra and are related to specific vibrations in cellular or extracellular components.  853 cm–1 (v(C–C) proteins), 1004 cm–1 (νs(C–C) ring breathing of phenylalanine), 1078 cm–1 (ν(C–C) of lipids), 1265 cm–1 (amide III v (C–N) and δ(N–H) of proteins), 1302 cm–1 (CH2 twisting and wagging of lipids), 1445 cm–1 (δ(CH2) deformation of proteins and lipids), 1618 cm–1 (v (C=C) of porphyrins), 1655 cm–1 (amide I v(C=O) of proteins), 2850 cm−1 and 2885 cm−1 (symmetric and asymmetric CH2 stretching of lipids), 2940 cm−1 (CH3 stretching of proteins), 3009 cm−1 (asymmetric = CH stretching of lipids). Bands above 3200 cm−1 due to water are not easily accessed in most instruments. Adenomas and adenocarcinomas which present prominent malignant latency, were related with significant reduced intensities of the lipid Raman peaks at 1078 cm–1 (v(C=C)), 1425 cm–1 (δ(CH2)) and 2850 and 2885 cm−1 (symmetric and asymmetric CH2 stretching) and marginally stronger signal at 1004 cm–1 due to νs (C–C) ring breathing of phenylalanine concomitant with up-regulated protein content.

Comment 3:   About the several Raman equipment specified for the specified bioapplication, the equipment details with figures should be presented.

Reply: Details of representative Raman techniques were added in lines 254-263 and lines 278-280. Moreover Figure 3 and 4 were added.

Lines 254-263: Moreover, novel Raman techniques combined with advanced fiber probes can provide a boost to Raman Spectroscopy’s application in clinical praxis. For example, Micro-scale spatially offset Raman spectroscopy with an optical fiber probe (micro-SORS) can collect photons from deeper layers by offsetting the position of laser excitation beam [82] reaching a penetration depth up to 5 cm [83] Recently, Zhang et al combined micro-SORS with Surface-enhanced Raman spectroscopy (SERS) applied on tissue phantom of agarose gel and biological tissue of porcine muscle [84] According to their results, the penetration depth could be improved over 4 cm in agarose gel and 5mm in porcine tissue compared to the depth of 2cm in agarose gel and 3mm in porcine muscle received by SERS measurements.

Lines 278-280: “Figure 4 shows the different geometries of fiber-optics probes used in clinical applications such as endoscopic probes without any focusing optics, confocal endoscopic probes, and fiber probes for side-viewing [89].

Comment 4: The disadvantage of Raman and the challenges towards real clinic usage should be discussed in more details.

Reply: The disadvantages of Raman spectroscopy were pointed out in lines 168-170, 304-309, and 309-311. However, a new separate paragraph was added in section challenges and future perspective in lines 389-423. The challenges and future perspectives were combined in a paragraph in order to cover other Reviewer’ comment.

  1. Challenges and future perspectives

The main advantages of RS, such as a) its non-invasive character and compatibility with tissue physiology due to the weak water signal, b) its suitability for in vivo fiber-optic applications on versatile cancers and c) its high specificity with simultaneous chemical analysis of the malignant tissues have been thoroughly described in the previous sections There are, however, important limitations of the technique which hinder its establishment in the clinical setting. The most prominent constraints are the following: a) the weak Raman signals which require long acquisition times, b) the strong autofluorescence background which affects the quality of the acquired spectra hiding the weak Raman features and complicates the analysis, c) the potential damage of the tissues by laser heating which is a rather complicated effect depending on the laser excitation wavelength and power, as well as the light absorption coefficient of the tissues and d) the subtle differences in the spectra which require sophisticated analysis [159]. Nevertheless, technical advancements in current generation Raman spectrometers and integration of machine learning techniques for big data analysis and cancer classification gradually tilt the balance towards the application of RS as a rapid diagnostic tool in clinical praxis. It is worthy to mention that most of the portable Raman spectrometer manufacturers make their own cooperative research in the field and include such applications in their technical notes. Despite the above limitations, Raman Spectroscopy constitutes a very promising technique for the in-situ cancer diagnosis. Since the achievement of adequate surgical margins is vital for disease control and survival, an intraoperative guidance tool such as Raman probes will significantly limit the subjective methods which surgical resection techniques are currently based on (such as palpation and visual inspection). However, the effort to reliably assess resection margins in surgical oncology also suffers from constrains inherent in oncological surgery, such as non standardised practices and impact of epithelial or other dysplacias at the margins, as well as differences in reporting the status of surgical margin. With regard to RS applications on excised tissues, the method employed for retrieval of sections from resection margins or bioptic samples and the issue of postresection tissue shrinkage introduce extra variability.

Reviewer 2 Report

The authors have reviewed the application of Raman spectroscopy in cancer diagnosis, surgery, and biopsy guidance. The review is well organized in background, Raman spectra analysis for tissue characterization and Advanced Raman systems in clinical praxis, which is a great guide for researchers and clinicians. I recommend it for the publication before minor revise.

  1. The caption in Figure 2 is incorrectly marked as Figure 1.
  2. Some format details need to be corrected, such as the unit of Raman shift needs superscript.
  3. The further perspective of this field is suggested to add in the end of the manuscript.

Author Response

STATEMENT OF CORRECTIONS

                                                 REVISED MANUSCRIPT

Dear Editor,

In response to the reviewers´ comments, the manuscript has been significantly revised. All the corrections have been performed according to the Reviewers’ suggestions. The changes have been highlighted with yellow color in the text. Bellow follows the replies and respective answers to the reviewers’ comments and enquiries. All answers have been uploaded separately as well.

Yours sincerely,

Dr. Spyratou Ellas

                              Reviewer’s comments and General Corrections

Reviewer 2

Comment 1: The caption in Figure 2 is incorrectly marked as Figure 1.

Reply: The caption in Figure 2 was corrected. Figure 2 was replaced by a more illustrative figure, according to the comment of Reviewer 3.   

Comment 2: Some format details need to be corrected, such as the unit of Raman shift needs superscript.

Reply:  Superscript was applied to all the units of the Raman peaks (cm-1)

Comment 3:  The further perspective of this field is suggested to add at the end of the manuscript.

Reply: A new section: “challenges and future perspectives” was added at the end of the manuscript in lines 389-423. The challenges and future perspectives were combined in a paragraph in order to satisfy other Reviewer’ comment.

  1. Challenges and future perspectives

The main advantages of RS, such as a) its non-invasive character and compatibility with tissue physiology due to the weak water signal, b) its suitability for in vivo fiber-optic applications on versatile cancers and c) its high specificity with simultaneous chemical analysis of the malignant tissues have been thoroughly described in the previous sections There are, however, important limitations of the technique which hinder its establishment in the clinical setting. The most prominent constraints are the following: a) the weak Raman signals which require long acquisition times, b) the strong autofluorescence background which affects the quality of the acquired spectra hiding the weak Raman features and complicates the analysis, c) the potential damage of the tissues by laser heating which is a rather complicated effect depending on the laser excitation wavelength and power, as well as the light absorption coefficient of the tissues and d) the subtle differences in the spectra which require sophisticated analysis [159]. Nevertheless, technical advancements in current generation Raman spectrometers and integration of machine learning techniques for big data analysis and cancer classification gradually tilt the balance towards the application of RS as a rapid diagnostic tool in clinical praxis. It is worthy to mention that most of the portable Raman spectrometer manufacturers make their own cooperative research in the field and include such applications in their technical notes. Despite the above limitations, Raman Spectroscopy constitutes a very promising technique for in-situ cancer diagnosis. Since the achievement of adequate surgical margins is vital for disease control and survival, an intraoperative guidance tool such as Raman probes will significantly limit the subjective methods which surgical resection techniques are currently based on (such as palpation and visual inspection). However, the effort to reliably assess resection margins in surgical oncology also suffers from constrains inherent in oncological surgery, such as non standardised practices and impact of epithelial or other dysplacias at the margins, as well as differences in reporting the status of surgical margin. With regard to RS applications on excised tissues, the method employed for retrieval of sections from resection margins or bioptic samples and the issue of postresection tissue shrinkage introduce extra variability.

Reviewer 3 Report

This manuscript reviews the advanced Raman spectroscopic techniques in clinical applications of cancer diagnosis and guided surgeries. The most of content is clear and the writing is fluent. I would recommend its publications after some revisions. See my comments below.

  1. In the first paragraphs of introduction part, the description of cancer is too verbose. The authors could shorten the first paragraph and try to combine the first/second paragraph into one.
  2. The title may be too wide. When first seeing the title, I thought the liquid biopsy using Raman spectroscopy would be included in this manuscript, since it was also related to cancer diagnosis and biopsy – but it turned out not. Because the manuscript is only focus on “in situ” detection at the tissue level, I would recommend the authors indicate this in title and abstract clearly.
  3. Also in the title, the authors classified the applications into three: diagnosis, surgery, and biopsy. While biopsy is obviously included in diagnosis, this classification is not accurate.
  4. There are two “Figure 1” s. The 2nd should be named as Figure 2.
  5. The information in Figure 2 has not been introduced clear enough. In the upper left corner of Figure 2, the machine learning model workflow should be explained. And what is the picture on the bottom left about? They should be introduced clearly either in the Figure title or in the context.
  6. When discussing the Raman fiber-optic probes, their better tissue penetration could be briefly discussed. See reference:

Optical penetration of surface-enhanced micro-scale spatial offset Raman spectroscopy in turbid gel and biological tissue, Journal of Innovative Optical Health Sciences, DOI: 10.1142/s1793545821410017

  1. Table 1 is wordy and unclarified. The key points, such as sensitivity/specificity, and the Raman devices used (i.e., single fiber probe, hand-held fiber-lens coupled probe, etc.) should be indicated in a SEPARATE column. Also, do not introduce several works in a single line – this is confusing, since it will be hard for the audience to locate the specific reference. I would suggest listing each reference in one line.
  2. There are some grammatical errors in the article. Page 2, line53: “the motivation to develop effective cancer diagnostic and treatment methods becomes strongly” should be “the motivation to … becomes strong”.

Author Response

STATEMENT OF CORRECTIONS

                                                 REVISED MANUSCRIPT

Dear Editor,

In response to the reviewers´ comments, the manuscript has been significantly revised. All the corrections have been performed according to the Reviewers’ suggestions. The changes have been highlighted with yellow color in the text. Bellow follows the replies and respective answers to the reviewers’ comments and enquiries. All answers have been uploaded separately as well.

Yours sincerely,

Dr. Spyratou Ellas

                              Reviewer’s comments and General Corrections

Reviewer 3

Comment 1: In the first paragraphs of the introduction part, the description of cancer is too verbose. The authors could shorten the first paragraph and try to combine the first/second paragraph into one.

Reply: The first paragraph was shortened and the first/second paragraphs were combined.

  1. Introduction

According to World Health Organization (WHO) [1] 14.1 million new cancer cases, 8.2 million cancer deaths, and 32.6 million people living with cancer were reported in 2012 [1]. in 2020 nearly 10 million cancer deaths have been accounted worldwide while the most common cancer cases pertain breast cancer (2.26 million cases); lung cancer (2.21 million cases); and colon and rectum cancer (1.93 million cases) [2]– [4]. Therefore, the early and accurate diagnosis as well as the precise and adequate surgical removal of a malignancy can led to the reduction of cancer’s high mortality rates [5], [6], in such a way as to surpass the standard methodology of histopathological examination based on the excision and biopsy of small portions of the lesional tissue [7]. Since the differentiation among benign tumours, premalignant, early-stage malignant, and healthy tissue is challenging, repeated biopsies are often necessary with. As it is stated in bibliography, for tissue sampling, the positive predictive values are as low as 22% for prostate cancer diagnosis, 1.4% for breast cancer, 18.5% in lung cancer screenings and 7–23% for melanoma diagnosis [8]-[11]. Various conventional imaging techniques have attempted to serve as adjuvant tools for biopsy and surgery guidance. In the field of ionizing radiation, positron emission tomography (PET), computed tomography (CT) and single photon emission computed tomography (SPECT) offer great results, with only undisputable drawback the dose deposition to the patient [12]-[14]. Simultaneously, magnetic resonance imaging (MRI), optical coherence tomography (OCT), white light reflectance (WLR), fluorescence and high frequency ultrasound and several other emerging spectroscopic techniques based on non-ionizing radiation have been proved valuable diagnostic tools avoiding burdening the clinical subject or the practitioner with extra dose [15]-[18]. Nonetheless, currently, no single imaging modality has been proven sufficient in terms of the required standards of specificity, sensitivity, multiplexing capacity, spatial and temporal resolution and low cost [19], [20]. Moreover, most techniques are unable to provide information regarding the molecular tissue composition [21], [22]. They just confide on visual changes of the tissue structure and thus present lack of specificity [23],[24]. 

Comment 2: The title may be too wide. When first seeing the title, I thought the liquid biopsy using Raman spectroscopy would be included in this manuscript, since it was also related to cancer diagnosis and biopsy – but it turned out not. Because the manuscript is only focus on “in situ” detection at the tissue level, I would recommend the authors indicate this in title and abstract clearly.

Reply:  The title was changed and the phrase “in situ” was added.

“Raman spectroscopy: a personalized decision-making tool on clinicians’ hands for in situ cancer diagnosis and surgery guidance”

Comment 3:  Also in the title, the authors classified the applications into three: diagnosis, surgery, and biopsy. While biopsy is obviously included in diagnosis, this classification is not accurate.

Reply:  The biopsy has been included in diagnosis and the text in “Abstract” and “Introduction” was formatted accordingly:

Abstract:

Lines 41-43: Nowadays, advanced Raman systems coupled with modern instrumentation devices and machine learning methods are entering the clinical arena as adjunct tools towards personalized and optimized efficacy in surgical oncology.”

Introduction:

Lines 113-115: “We highlight the perspective of the application of advanced Raman systems in clinical praxis as an adjunct tool towards early diagnosis, biopsy guidance and oncologic surgery guidance.”

Comment 4:  There are two “Figure 1” s. The 2nd should be named as Figure 2.

Reply:  The caption in Figure 2 was corrected. Figure 2 was replaced by a more illustrative figure, according to the comment 5.   

Comment 5: The information in Figure 2 has not been introduced clear enough. In the upper left corner of Figure 2, the machine learning model workflow should be explained. And what is the picture on the bottom left about? They should be introduced clearly either in the Figure title or in the context.

Reply:  Figure 2 was replaced by a new one which clearly explains the machine learning model workflow.

Lines 177-178: “Figure 2 is a schematic representation of the workflow of the combination of Raman spectroscopy with machine learning models for tissue discrimination and classification using a multilayer perceptron (MLP).”

Legend of Figure 2: “Figure 2. Depicts the basic structure of Machine Learning workflow applied on a Raman Dataset.”

Moreover, we modified the text regarding artificial intelligence to separate section: “3. Machine Learning and Deep Learning as tools towards Raman spectra analysis” was added. In this section the following sentences were added:

Lines 181-189: In more detail, features extracted from Raman spectra (e.g.  representative intensities at certain wavenumber) and/or from spectral images of biostructures (e.g. pixel intensity patterns) are used as inputs to a MLP. The hidden layers of a MLP will introduce a series of linear and non-linear calculations that will lead to a single output neuron. Each output corresponds to a normal or tumour class for discrimination between healthy and cancerous tissues. Several others machine learning models such as Support Vector Machine (SVM) [65], Decision trees (DTs) boosted tress [66], convolutional neural networks (CNNs)[67] and or Artificial neural networks (ANNs) [68] have been exploited against cancer detection for almost 20 years69].

Lines 225-231: “Non-linear NNs have been used to predict the Bayesian probability of breast cancer. Nine spectra regions, six in the fingerprint region (600-1800 cm-1) and three in the high wavenumber region (2800-3200 cm-1) were identifying comparing DNA/RNA, protein, carbohydrate and lipid cellular components of health and cancer cells [74]. Deep convolution neural networks have been applied to fiber optic Raman spectroscopy systems providing a novel classification method for tongue squamous cell discrimination [75] According to the result, high sensitivity (99.31%) and specificity (94.44%) were achieved.

Comment 6:  When discussing the Raman fiber-optic probes, their better tissue penetration could be briefly discussed. See reference:

Optical penetration of surface-enhanced micro-scale spatial offset Raman spectroscopy in turbid gel and biological tissue, Journal of Innovative Optical Health Sciences, DOI: 10.1142/s1793545821410017

Reply: The Raman fiber-optic probes and their better tissue penetration were briefly discussed and the suggested reference was incorporated. Moreover, an extra figure (Figure 4) was added.

Lines 247-263: “Advanced fiber probes such as probes with plasmonic nanostructures on their distal end surface can provide the enhancement of the surface Raman scattering signal [80] Moreover, fiber probes can overcome the limited penetration depth of laser radiation in tissues due to the high diffusion and scattering of photons. Figure 3 shows a portable Raman imaging system based on SERS fiber- optics probes capable of conducting white light endoscopy [81]. Moreover, novel Raman techniques combined with advanced fiber probes can provide a boost to Raman Spectroscopy’s application in clinical praxis. For example, Micro-scale spatially offset Raman spectroscopy with an optical fiber probe (micro-SORS) can collect photons from deeper layers by offsetting the position of laser excitation beam [82] reaching a penetration dept up to 5 cm [83] Recently, Zhang et al combined micro-SORS with Surface-enhanced Raman spectroscopy (SERS) applied on tissue phantom of agarose gel and biological tissue of porcine muscle [84] According to their results, the penetration depth could be improved over 4 cm in agarose gel and 5mm in porcine tissue compared to the depth of 2cm in agarose gel and 3mm in porcine muscle received by SERS measurements.

Lines 278-281: “Figure 4 shows the different geometries of fiber-optics probes used in clinical applications such as endoscopic probes without any focusing optics, confocal endoscopic probes and fiber probes for side-viewing [89].  

Legend of Figure 4: Different geometries of fiber-probes used in clinical applications: (a) nonsuperficial endoscopic probe with one excitation fiber in the center and seven collection fibers arranged around the emitter (b) confocal endoscopic fiber probe with a ball lens (c) fiber probes with mirror (or prisms) [89,] doi: 10.1117/1.JBO.23.7.071210. PMID: 29956506 -Excitation and collection filters are also depicted.

Comment 7: Table 1 is wordy and unclarified. The key points, such as sensitivity/specificity, and the Raman devices used (i.e., single fiber probe, hand-held fiber-lens coupled probe, etc.) should be indicated in a SEPARATE column. Also, do not introduce several works in a single line – this is confusing, since it will be hard for the audience to locate the specific reference. I would suggest listing each reference in one line.

Reply: The table was formatted according to the Reviewer’s instructions.

Comment 8: There are some grammatical errors in the article. Page 2, line53: “the motivation to develop effective cancer diagnostic and treatment methods becomes strongly” should be “the motivation to … becomes strong”.

Reply: The whole text was checked by a native English-speaking colleague. Any syntax or grammatical errors were corrected.

The sentences in line 53 was deleted as the whole paragraph was reduced according to comment 1. 

Reviewer 4 Report

The manuscript is intended as a review of Raman spectroscopic applications for use as a diagnostic tool for rapid and non-radiative discrimination between cancerous and non-cancerous human tissue. The authors demonstrate a vast knowledge of the subject by including a large number of citations while not appearing to favor only a select few. I, however, am not an expert in the analysis of human tissue, cancerous or not, but much more knowledgeable in the uses of Raman spectroscopy. I, therefore, limit my comments and insist that the editorial staff include other referees much more capable in the specific field of tissue analysis. My overall opinion of the manuscript is that it requires major revision. I believe that the aims of the paper are straightforward, as it is essentially a review regarding a specific topic and I believe it merits attention as a promising application at the clinical level. The paper, however, suffers from poor overall organization and occasional areas that are not well elaborated. Simply deconstructing the paper into an outline of its principal parts shows a lack of linearity and an inability to maintain focus. Some figures and tables are unacceptable in their present form. See my specific comments that follow.

1. L86-87 unclear what is intended by "biostructure". Raman spectroscopy is considered a short-range order (SRO) spectroscopic technique. Unlike x-ray diffraction, as an example of a long-range order method, bands in the Raman spectra result from the presence of molecular species or polyatomic groups typically on the order of only a few angstroms in diameter. For example, a carbonyl group attached to a large organic compound will show a peak in the Raman spectrum very close to 1740 cm-1, regardless of the structure of the molecule much further from the carbonyl group. Its exact frequency will be much more dependent on the local structure of the molecule closest to the carbonyl group. Therefore, the term biostructure is vague.

2. L152-164 As a review article, I would expect some expansion on the topics of combining artificial intelligence techniques with Raman spectroscopy within this field. As an in vivo clinical tool, these appear to be fundamental issues apart from the specific applicability of Raman for distinguishing different types of tissues. In this paragraph, there is simply a listing of about ten citations before the article continues on to the following paragraph regarding probability and error analysis. 

3. L178-199 This is more or less how a single paragraph in a review article should appear. The major spectral indicators are identified. Methods of data analysis are indicated. Results are summarized succinctly. It is important that the sequence of paragraphs such as these are organized to best facilitate the aims of the review. 

4. L203 This is labelled as Fig. 1 where it should be Fig. 2. The figure itself is not very useful to the reader in my opinion as frequency scale is absent from the Raman spectra, fields in the flow chart are left blank and neither color or size scale is included in the image. The paragraph beginning at L200 abruptly ends at L201 in the text making the figure even less useful. 

5. L207-265 Section 3.1 is better written compared to the earlier sections. As a paper to be consumed mainly by those in medical research I can understand the abrupt topical switch from section 2 to section 3 but it almost seems contrary to a "normal" review paper that might first touch upon technical development before seeing those applied to the clinical environment. I question the authors motivation for organizing the paper in this way. 

6. L286-292 Similar to an earlier comment there could be some expansion of discussion here. The following two paragraphs (L293-302, L303-319) are much better examples of more efficient writing.

7. Table 1 is absolutely unacceptable. It runs over 6 pages of the manuscript and does not allow the reader to locate specific information rapidly due to the poor organization of the individual cells. I would suggest that the table is condensed and possibly divided into multiple tables based on cancer type, for example. The individual cells are filled with too much text. The authors should develop a more intelligent way to deliver data relevant information and avoid wordy descriptive text. 

Author Response

STATEMENT OF CORRECTIONS

                                                 REVISED MANUSCRIPT

Dear Editor,

In response to the reviewers´ comments, the manuscript has been significantly revised. All the corrections have been performed according to the Reviewers’ suggestions. The changes have been highlighted with yellow color in the text. Bellow follows the replies and respective answers to the reviewers’ comments and enquiries. All answers have been uploaded separately as well.

Yours sincerely,

Dr. Spyratou Ellas

                              Reviewer’s comments and General Corrections

Reviewer 4

Comment 1: L86-87 unclear what is intended by "biostructure". Raman spectroscopy is considered a short-range order (SRO) spectroscopic technique. Unlike x-ray diffraction, as an example of a long-range order method, bands in the Raman spectra result from the presence of molecular species or polyatomic groups typically on the order of only a few angstroms in diameter. For example, a carbonyl group attached to a large organic compound will show a peak in the Raman spectrum very close to 1740 cm-1, regardless of the structure of the molecule much further from the carbonyl group. Its exact frequency will be much more dependent on the local structure of the molecule closest to the carbonyl group. Therefore, the term biostructure is vague.

Reply: The sentence in L 86-87 (lines 78-83 in the revised version of the manuscript) was deleted. A new explanatory sentence was also added below

Lines 78-83: The spectral fingerprint which is produced during laser-biomolecule interaction is unique and characterizes the biomolecule. RS provides information on the short-range molecular vibrational level. Thus, the Raman bands are characteristic of the molecular bonding in the chemical groups, whereas chemical conformation and environment at macromolecular level determine the exact frequencies of the Raman bands.

Comment 2: L152-164 As a review article, I would expect some expansion on the topics of combining artificial intelligence techniques with Raman spectroscopy within this field. As an in vivo clinical tool, these appear to be fundamental issues apart from the specific applicability of Raman for distinguishing different types of tissues. In this paragraph, there is simply a listing of about ten citations before the article continues on to the following paragraph regarding probability and error analysis.

Reply:  We modified the text regarding artificial intelligence to a separate section: “3. Machine Learning and Deep Learning as tools towards Raman spectra analysis” was added. In this section the following sentences were added:

Lines 181-189: “In more detail, features extracted from Raman spectrums (e.g.  representative intensities at certain wavenumber) and/or from spectral images of biostructures (e.g. pixel intensity patterns) are used as inputs to a MLP. The hidden layers of a MLP will introduce a series of linear and non-linear calculations that will lead to a single output neuron. Each output corresponds to a normal or tumour class for discrimination between healthy and cancerous tissues. Several others machine learning models such as Support Vector Machine (SVM) [65], Decision trees (DTs) boosted tress [66], convolutional neural networks (CNNs) [67] and or Artificial neural networks (ANNs) [68] have been exploited against cancer detection for almost 20 years69].  

Lines 225-231: Non-linear NNs have been used to predict the Bayesian probability of breast cancer. Nine spectra regions, six in the fingerprint region (600-1800 cm-1) and three in the high wavenumber region (2800-3200 cm-1) were identifying comparing DNA/RNA, protein, carbohydrate and lipid cellular components of health and cancer cells [74]. Deep convolution neural networks have been applied to fiber optic Raman spectroscopy systems providing a novel classification method for tongue squamous cell discrimination [75] According to the result, high sensitivity (99.31%) and specificity (94.44%) were achieved.

Comment 3:  L178-199 This is more or less how a single paragraph in a review article should appear. The major spectral indicators are identified. Methods of data analysis are indicated. Results are summarized succinctly. It is important that the sequence of paragraphs such as these are organized to best facilitate the aims of the review.

Reply: Based on the appropriate format of this paragraph (lines 190-231 in the revised version of the manuscript) this specific paragraph and more paragraphs in the manuscript were altered accordingly.

Lines 134-136: “Prominent tissue Raman peaks are observed in the fingerprint, 800-1800 cm-1 and the high frequency, 2800-3200 cm-1, spectral regions. As a characteristic example we present Raman spectra from colorectal tissues by Bergholt et al”.

Lines: 140-153: “The strongest Raman bands are marked upon the spectra and are related to specific vibrations in cellular or extracellular components.  853 cm–1 (v(C–C) proteins), 1004 cm–1 (νs(C–C) ring breathing of phenylalanine), 1078 cm–1 (ν(C–C) of lipids), 1265 cm–1 (amide III v (C–N) and δ(N–H) of proteins), 1302 cm–1 (CH2 twisting and wagging of lipids), 1445 cm–1 (δ(CH2) deformation of proteins and lipids), 1618 cm–1 (v (C=C) of porphyrins), 1655 cm–1 (amide I v(C=O) of proteins), 2850 cm−1 and 2885 cm−1 (symmetric and asymmetric CH2 stretching of lipids), 2940 cm−1 (CH3 stretching of proteins), 3009 cm−1 (asymmetric = CH stretching of lipids). Bands above 3200 cm−1 due to water are not easily accessed in most instruments. Adenomas and adenocarcinomas which present prominent malignant latency, were related with significant reduced intensities of the lipid Raman peaks at 1078 cm–1 (v(C=C)), 1425 cm–1 (δ(CH2)) and 2850 and 2885 cm−1 (symmetric and asymmetric CH2 stretching) and marginally stronger signal at 1004 cm–1 due to νs (C–C) ring breathing of phenylalanine concomitant with up-regulated protein content.

Lines 225-231: Non-linear NNs have been used to predict the Bayesian probability of breast cancer. Nine spectra regions, six in the fingerprint region (600-1800 cm-1) and three in the high wavenumber region (2800-3200 cm-1) were identifying comparing DNA/RNA, protein, carbohydrate and lipid cellular components of health and cancer cells [74]. Deep convolution neural networks have been applied to fiber optic Raman spectroscopy systems providing a novel classification method for tongue squamous cell discrimination [75] According to the result, high sensitivity (99.31%) and specificity (94.44%) were achieved.

Lines 348-354: An intraoperative Raman system that measures directly brain tissue in the patient, has proven to distinguish dense and low-density cancer infiltration from benign brain tissue with a sensitivity of 93% and a specificity of 91% [105],[106].  More precisely, the experimental setup was pertaining a hand-held optic Raman probe and a 785nm NIR Laser [106]. The research team exploited the boosted trees supervised machine learning algorithm to process their data and eventually differentiate the spectrum among cancerous and healthy brain tissue [106].

Lines 354-359: In another study, a real-time Raman intraoperative system was used during breast cancer surgery for assessment of freshly resected specimens [107]. A total of 220 Raman spectra were collected with the aid an 830-nm-diode laser focused on a Raman optical fiber probe [107]. This study has demonstrated that Raman spectroscopy could discriminate cancerous tissues from normal breast tissues with a sensitivity of 83% and a specificity of 93% [107].

Comment 4:  L203 This is labelled as Fig. 1 where it should be Fig. 2. The figure itself is not very useful to the reader in my opinion as frequency scale is absent from the Raman spectra, fields in the flow chart are left blank and neither color or size scale is included in the image. The paragraph beginning at L200 abruptly ends at L201 in the text making the figure even less useful.

Reply: The figure caption was corrected. Fig.2 was replaced by a new one which explain clearly the machine learning model workflow.

Lines 177-179: “Figure 2 is a schematic representation of the workflow of the combination of Raman spectroscopy with machine learning models for tissue discrimination and classification using a multilayer perceptron (MLP).”

Legend of Figure 2: “Figure 2. Depicts the basic structure of Machine Learning workflow applied on a Raman Dataset.”

Comment 5: L207-265 Section 3.1 is better written compared to the earlier sections. As a paper to be consumed mainly by those in medical research I can understand the abrupt topical switch from section 2 to section 3 but it almost seems contrary to a "normal" review paper that might first touch upon technical development before seeing those applied to the clinical environment. I question the authors motivation for organizing the paper in this way.

Reply: The topical transition from section 2 to section 3 became more continuous through the insertion of additional sections (section 3. Machine Learning and Deep Learning as tools towards Raman spectra analysis was added and section 3 became section 4. Advanced Raman systems in clinical praxis in the revised manuscript). As can be depicted in previous comment’s replies, the insertion of additional paragraphs and the rearrangement of the manuscript attempted to constitute it more well organized for the reader.

Comment 6: L286-292 Similar to an earlier comment there could be some expansion of discussion here. The following two paragraphs (L293-302, L303-319) are much better examples of more efficient writing.

Reply: Focused on the appropriate format of previous L178-199 (current lines 190-231), L286-292 were enriched.

Lines 348-354: An intraoperative Raman system that measures directly brain tissue in the patient, has proven to distinguish dense and low-density cancer infiltration from benign brain tissue with a sensitivity of 93% and a specificity of 91% [105],[106].  More precisely, the experimental setup was pertaining a hand-held optic Raman probe and a 785nm NIR Laser [106]. The research team exploited the boosted trees supervised machine learning algorithm to process their data and eventually differentiate the spectrum among cancerous and healthy brain tissue [106].

Lines 354-359: In another study, a real-time Raman intraoperative system was used during breast cancer surgery for assessment of freshly resected specimens [107]. A total of 220 Raman spectra were collected with the aid an 830-nm-diode laser focused on a Raman optical fiber probe [107]. This study has demonstrated that Raman spectroscopy could discriminate cancerous tissues from normal breast tissues with a sensitivity of 83% and a specificity of 93% [107]. A handheld contact Raman spectroscopy probe was used for real-time identification of brain cancer during the surgery. Jermyn et al. obtained very fast and high-quality pure Raman signals from 0.5 mm tissue areas with sampling depth up to 1mm during the tumour resection [105] by using micrometer-scale filters that were placed directly at the tip of the optical fibers [105].

Comment 7: Table 1 is absolutely unacceptable. It runs over 6 pages of the manuscript and does not allow the reader to locate specific information rapidly due to the poor organization of the individual cells. I would suggest that the table is condensed and possibly divided into multiple tables based on cancer type, for example. The individual cells are filled with too much text. The authors should develop a more intelligent way to deliver data relevant information and avoid wordy descriptive text.

Reply: The table was formatted according to the Reviewer’s instructions/suggestions.
